# Paternal Leave Entitlement and Workplace Culture: A Key Challenge to Paternal Mental Health

**DOI:** 10.3390/ijerph20085454

**Published:** 2023-04-10

**Authors:** Ernestine Gheyoh Ndzi, Amy Holmes

**Affiliations:** York Business School, York St. John University, York YO31 7EX, UK; a.holmes@yorksj.ac.uk

**Keywords:** paternity leave, shared parental leave, fathers’ mental health, fatherhood, parenthood, COVID-19, wellbeing

## Abstract

Paternal mental health continues to be a health concern in the UK. Paternal leave entitlement and workplace cultures have failed to support fathers in navigating the complexity of fatherhood, which has an impact on fathers’ wellbeing. Interviewing twenty fathers in the York area, this study seeks to explore the impact of parental leave entitlements and workplace cultures on fathers’ mental health. The findings demonstrate that the influence of gendered norms and hegemonic masculinity perceptions are ingrained in the current leave entitlement and workplace cultures. While fathers are entitled to take leave, the leave is significantly insufficient to allow them to forge a meaningful bond with a newborn or adapt to the change in routine brought about by the birth of a baby. Furthermore, workplace cultures fail to recognise the responsibilities that come with fatherhood and provide insufficient support for fathers. The COVID-19 lockdown presented fathers with a unique opportunity to be available and take on more family responsibilities. Fathers felt they did not have to navigate gendered and hegemonic perceptions to spend more time with the family. This paper challenges structural and cultural barriers that prevent fathers from taking leave and impacting negatively on fathers’ mental health. The paper suggests a review of the current paternal leave entitlement and cultural change in the workplace.

## 1. Introduction

Mental health is a contemporary health and policy concern, with a fifth of men (19.5%) presumed to receive a mental health related diagnosis during adulthood [1]. It is important to understand the factors that influence male mental health and wellbeing in the context of fatherhood, particularly working fathers.

Traditional gender norms emphasise the father’s role as the breadwinner in typical family structures. An expanding body of research continue to consider the role of traditional gender norms in the persistence of gender inequalities in earnings [2], labour force participation [3], and domestic roles [4]. Although attitudes toward perceived gender differences in parenting persist [5,6], the modern fatherhood ideal encourages increased and active involvement of fathers in their children’s lives [1]. The new fatherhood ideals encourage fathers to be affectionate and nurturing [7]. Men who strongly identify with the new fatherhood ideal may deemphasize, reject, or reshape traditional masculinity to take on contemporary fathering roles [8]. Petts et al. [9] found a negative association between adherence to masculine norms and endorsement of the new fatherhood ideal. Masculinity is the concept of men’s identity as the breadwinner and ‘master of the house’ in normative heterosexual nuclear family structures [10]. This concept operates differently both within and beyond the home. For fathers seeking to balance work- and home-life commitments, cultural representations of hegemonic masculinity presenting the ideal worker as separate from emotional and family responsibilities can be particularly challenging [11]. Masculine identities are often constructed in relation to their working environment [12]. Traditional gender norms [13] emphasise the father’s role as the breadwinner in typical family structures. Hegemonic discourses of masculinity produce and reinforce assumptions about men’s role as parents [14]. A perceived lack of support at an organisational level has been identified as a negative predictor of paternal leave-taking [15,16]. Workplace norms characterised by the ideal unencumbered worker amplify the expectation that men are indispensable to the workplace [17].

Mothers and fathers navigate structurally distinct pathways into parenthood, largely mediated by a gendered division of paid and unpaid labour [18] and consolidated by differential experiences of parental leave [19]. Although the UK government has mandated paid paternity leave which offers a promising social intervention for de-gendering the division of labour in domestic spheres [20], the disproportionality of leave entitlement between a father (two weeks) and mother (fifty-two weeks) suggests that fathers are considered as the breadwinner in the family with little parental responsibilities when the baby is born. A growing body of research conceptualises men as active co-parents with equitable nurturing capabilities to mothers [21], as opposed to forcing fathers into a peripheral role as the mother’s helper/provider [22]. Such research emphasises sharing parenting tasks from the beginning to instil greater confidence and skill in a father’s perceived parenting [23] which sufficient paternity leave infrastructure could facilitate [24,25]. Furthermore, children with higher paternal involvement are reported to be healthier [26], happier [27], present with fewer behavioural problems, and perform better in school [28].

Parental leave for fathers has been introduced in several countries [29], increasing the proportion of fathers who have become more involved, nurturing, and engaged with their children [30]. Through taking paternal leave, fathers are offered an increased opportunity to invest time in taking sole responsibility for their children. This is particularly of interest since studies have identified qualitative differences between the time fathers spend with the child alone vs. when the mother is also present [30]. Caring alone strengthens the father–child bond at a greater rate, facilitating the acquisition of new parenting skills in the father, which enhances their self-esteem [31]. Enhanced sensitivity in responding to a child’s needs and greater levels of confidence are reported in fathers who spend time alone parenting an infant [32].

Homogenous legally mandated parental leave policies are one of the most efficient solutions to standardising the irregularity in parental leave-taking arrangements [33]. A concurrent lack of discussion about fathers’ leave-taking decisions [34] and absence of ‘managerial champions’ [35] laying the cornerstones for these conversations manifest in a scarcity of social support for fathers at work [19,36]. The decision of whether to opt-in to parental leave is considered a ‘right, but not a duty’ by some fathers [37]. Parallel to the literature on maternity leave judgements [38], paternity leave is reduced to a choice at the level of an individual [39]. The use of leave by fathers has been found to decrease risk of depression [40], stress [41], and improve mental health [42]. Extended parental leave has been found to further decrease stress and anxiety in fathers [43,44]. Research demonstrates that when fathers take leave, their availability and support improve the mental health of their partners [45] and benefits the child by encouraging father–child bonding after birth and enabling commitments to fathers’ engagement [46].

Despite being marketed as ‘gender-neutral’ in its policy aims, UK Shared Parental Leave (SPL) legislation prioritises the mother’s caregiving role [47], inherently situating men as an adjunct to the mother’s dominant caregiving role [48]. In 2019, it was estimated that a mere 2% of fathers in the UK opted to take SPL [49], with the financial consequences of temporarily exiting the labour market largely governing these decisions [39]. The low level of SPL pay does not offer great enough economic compensation to persuade fathers to take time away from work. A common pattern amongst fathers in the UK is to opt for one week paid paternity leave and an additional week taken as paid holiday [50]. Economic provisions are used by fathers in rationalising disparate leave choices, whereby it is framed as natural for the partner with a lower salary to take longer leave [51]. Furthermore, fathers emphasise finances as a key determinant on their leave-taking decision-making [39], which reinforces the gender stereotype of fathers being the breadwinners of the family. The existence of the gender pay gap gives the impression that mothers will always take longer maternity leave because of the differences in pay and places little emphasis on paternal bonding and mental health [39].

Considerations on behalf of the mother are salient in the decision-making process of taking paternity leave [52]. In the context of shared parental leave (SPL), fathers tend to grant ownership to the mother rather than treating it as a collective leave benefit entitlement [53]. Negotiations within a partnership are judged to be ‘mother-led’ from a father’s perspective [54], with breastfeeding heavily dictating the distribution of leave time [55]. Gendered parenting ideology relates ‘good mothering’ to intensive embodiment of caregiving duties (i.e., breastfeeding) [56,57], which opposes ‘good fathering’ (i.e., providing stability via financial income) [58,59]. Fathers’ reluctance to ‘take-away’ from mothers’ leave is a recurrent theme in the literature [39,54], underpinned by physiologically focused discourse [60] framing women as natural caregiving experts [61], signposting how paternal leave is positioned as a mother’s right, rather than an equal right.

The COVID-19 lockdown created an opportunity for fathers to be physically present at home, which resulted in more engagement in family activities [62]. There was a positive increase in shared parenting responsibilities, although lack of access to informal support proved challenging and impacted on fathers’ mental health [63]. While it has been demonstrated that the COVID-19 lockdown had a negative impact on paternal experience and father–baby bonding as a result of fathers being excluded from maternity care, it is essential to explore fathers’ experiences of leave entitlement post-birth [64]. While these studies are insightful, it is essential to understand fathers’ experiences of leave entitlement and occupation cultures impact before and during the COVID-19 lockdown period.

This paper explores the impact of fathers’ leave entitlement and what role occupational cultures play in fathers’ mental health. Adopting semi-structured interviews with twenty fathers in the York area, the study explores the barriers that leave entitlement and occupational cultures create for fathers as they try to balance work and fatherhood. Although many families adopt the modern family ideals [5], masculinity and gendered division of labour perceptions are deeply rooted in leave policies, with a significant impact on fatherhood and fathers’ mental health. The paper demonstrates that current UK parental leave and occupational cultures play a key part in increasing perinatal mental health.

The study will also explore the experiences of some of the fathers whose babies were born during the COVID-19 lockdown in the UK where most people had to work from home or were furloughed.

## 2. Materials and Methods

Data for the study were collected through semi-structured interviews with twenty fathers in the York area. The call for participants was advertised on social media platforms, such as Twitter, Facebook, and Instagram, using snowballing and convenient sampling [65]. All interviews were conducted online via Zoom during the periods of April and June 2021, lasting on average fifty minutes. Online interviewing was the only viable means of collecting data because of the COVID-19 social interaction restrictions.

Participants that took part in the research were employed and working in areas such as education, the army, the police, healthcare, and administration. Most of the participants (15) identified themselves primarily as ‘White British’, 2 as ‘Black British’, and 3 participants did not disclose their ethnic backgrounds. Five of the participants became first-time fathers during the COVID-19 lockdown period. A further five of the fathers had their second babies born during the COVID-19 lockdown period. It is important to highlight this because their experiences, particularly for first-time fathers, might be different.

Data for the study were collected through interviews which consisted of six open-ended questions intended to cover four key areas—(1) fathers’ leave entitlement, (2) occupational cultures and influence on fathers’ leave attitude, (3) the financial challenges of fathers taking leave; and (4) COVID-19 lockdown impact.

The interview sought to understand how fathers’ leave entitlement and occupational cultures impacts on fathers’ mental health, and also capture the experiences of some of the fathers whose babies were born around the COVID-19 pandemic period. To better understand the relationship between paternal mental health, a thematic analysis of the transcripts was conducted using the NVivo software, sourced from the University’s website on data analysis tools to establish and code emergent themes. The method allowed the researcher to identify, analyse, and report codes and themes from a qualitative data set [66], in this case from interview transcripts. This was important to ascertain the impact of fatherhood on paternal mental health, with access to leave, workplace culture, and financial cost emerging as the most pertinent avenue for exploration in the context of this paper. Discussing fathers’ leave entitlement and workplace cultures allowed for better understanding of how men perceive their role as a parent in relation to work and the impact on paternal wellbeing.

While the study makes a valuable contribution to the understanding of paternal mental health studies, it is important to note that data were collected between April and June 2021, offering a different view of some fathers’ experiences during the pandemic. Findings of the study cannot be generalised across regions or socio-economic groups because participants were mainly from the York area. However, the leave entitlement for fathers is not unique to these fathers but is applicable to all fathers in the UK.

## 3. Results

The findings indicate that fathers’ decisions to take leave are influenced by legal, organisational, and economic factors. Analysis of the data revealed four key themes that influenced the participants’ perceptions and decision-making processes of taking parental leave: (1) fathers’ leave entitlement, (2) occupational cultures, (3) financial barriers to fathers taking parental leave, and (4) the impact of COVID-19. Table 1 below, illustrates how the themes were put together.

The exploration of these themes indicates the legal and sociocultural perceptions of fatherhood, which are influenced by leave entitlement, information, and support available to fathers. Exploration of these themes indicates that legal and occupational perceptions of fatherhood influence the information, support, type, and length of leave that is offered to fathers. Furthermore, participants’ internalised interpretations of hegemonic ‘breadwinning’ roles impact their decisions to take leave beyond financial motivating factors. The strain of reconciling these roles, and of navigating economic and occupational factors, impacts the participants’ mental health, with many feeling that they are neglected or punished as working fathers. However, this is not universal to all workplaces, and where participants felt supported to take leave, they reported more positive experiences. This illustrates that organisational support is vital for many fathers, as it significantly impacts not only their wellbeing, but that of their family in the aftermath of leave-taking decisions.

### 3.1. Leave Entitlement

While maternity well is well established in the UK and generous in terms of the period allowed for a mother to be on maternity leave, paternity leave is less generous, restricted to two weeks and SPL being an option to fathers only if mothers consent to sharing their maternity leave. The current paternity leave and SPL provisions for fathers highlight the gendered division of labour and ideas of masculinity around parenting. Fathers in the study expressed frustration over the length of paternity leave as being too short. Participant H expressed his inability to support his wife as needed and to bond with the baby because two weeks was too short:


*“Two weeks just isn’t enough at all. I mean my wife absolutely struggled like anything you know, and I just think we just needed a bit longer. And I think for me as well to be able to bond with my son I needed longer than two weeks, it’s not long enough, no.”*
(Participant H)

Participant I expressed concerns of being unable to adjust to the change and having to return to work, “I don’t think two weeks is long enough personally to adjust to the changes of having a newborn and then having to get back into work again.”

Participant N explained how the struggle to balance work and family life affected his mental health when the baby was born.


*“My mental health has definitely improved. So, I definitely got to a point where I was finding things very difficult. And I was thinking can I do this; can I take time off and be present for him?”*
(Participant N)

Fathers find the idea of sharing the mother’s leave in the form of SPL intrusive and would prefer to have an independent leave. This demonstrates the role hegemonic masculinity plays in dissuading fathers from taking leave even where the mother is willing to share. Participant E talked about not wanting to take the time away from his wife that he considered she deserved: ‘If we share it means that my wife will not be having more than, or having what she deserves, by effectively sharing it with her.’

Where the family had to spend more time in the hospital depending on the circumstances after birth, participants felt that the two weeks was further reduced as they spent time going back and forth from the hospital with little time left to spend with the baby and mother when they returned home, as expressed by Participant M:


*“The basic two weeks, you stay up and that, and you’re not really bonding with the child, plus they were still in hospital for like four days. And my paternity leave had already started. So, I spent more time going back and fore to the hospital.”*


Participant E commented that a longer paternity leave would be helpful to fathers and the child. He further commented on the undue pressure placed on mothers following the end of the fathers’ short two weeks paternity leave, as the mother is expected to carry on at home unsupported:


*”I think that if fathers were allowed more time to spend time with the newborn it would be more helpful, not just to themselves but to the kid as well. I think by the fathers not having enough time off work to help in the house, then there is more pressure on the women to effectively carry on from the pregnancy to childbirth to bringing up the child to the next thing, because the father is never going to be there as often as they would like to.”*


### 3.2. Occupational and Cultural Perceptions

Throughout the sample, the participants conveyed that their experience of leave was frequently informed by the attitudes of their employers and colleagues, and the organisational structures of the workplace. Although statutory leave is a protected right, some of the participants still felt resistance from their employers, and Participant K expressed relief that statutory paternity leave is ring-fenced in law:

*“She [K’s line manager] inhaled, “Oh I don’t know if we’re going to be able to facilitate that.” And it was hugely rewarding to have to say, “It’s okay, it’s statutory, you don’t have a choice.” I think that was, it was just a relief to know that my- I don’t know what it’s like with other employers but with schools a head teacher has all the power.”*
(Participant K)

Participant K’s response demonstrates that some employers see paternity leave as discretionary, rather than as a right that is equivalent to maternity leave. Therefore, statutory provisions protect fathers in the immediate period after childbirth; it is intimated here that without the legal precedent, he would have been denied leave. The attitude of his employer is illustrative of a broader cultural environment in which some employers do not afford parity to their male employees, and instead treat paternity leave as either a luxury or a nuisance.

Furthermore, fathers throughout the sample indicated that outside of statutory leave, additional time off is perceived as an encroachment upon their workplace responsibilities. Their colleagues and bosses expressed resentment towards them for taking extended periods of time off or for shifting to flexible working. For Participant E, the attempt to take additional time away from work were met punitively, with the threat of dismissal as a tangible motivator for returning to work. The sense that “[he] can become a part time employer or find something that will be suitable for [him] and [his] family”, is illustrative of the opposition between workplace’s culture and the needs of his family. This occupational rigidity, and the choice between “[going] back to work full-time or [being] laid off” (E) meant that he only took the statutory period due to fears that he would be sanctioned.

The study indicates that this is not uncommon, and that many fathers in the sample felt discouraged by their workplaces’ attitudes towards leave. Participant J reflects that that rather than dissuade their workers from taking extended leave, he felt that his employers would actively prevent them from doing so:

*“Another person was insistent he was going to have shared leave and my employers sort of tried to block it as much as they possibly could.”*
(Participant J)

Similarly, he feels that SPL “wasn’t even put on the table”, and that when he worked from home, he was treated with scrutiny for doing so:


*“the kind of attitude was sort of like, “Well, you’ve got your two weeks but you can’t take more than your two weeks. What are you doing being at home”? Even though I was working and on the call, “You can’t be at home.”*


Therefore, even when he worked remotely, Participant J felt that he was subjected to undue scrutiny for not being present in the workplace. Here, it is indicated that fatherhood is not perceived to be equal to motherhood in terms of its impact on the individual, and that many organisations tacitly reinforce the gender binary in relation to parental roles. Even where participants took unpaid leave, some colleagues reinforced the expectation that they worked whilst they were at home, thus consolidating gendered perceptions of parental leave:


*“...Even as I was going on that period of leave, a couple of people were saying, “But you’ll be checking emails and things while you’re out, won’t you?” And I said, “Well, no, I mean, one, you’re not paying me for this, despite, as we’ve discussed a number of times, and you’ve refused to do, so I’m on unpaid leave. And would you be asking a mum who’s about to go on mat leave, you will be checking your emails won’t you?”*
(Participant A)

As indicated by Participant A, many fathers are still expected to prioritise the workplace by performing administrative tasks, such as checking emails. He reflected that in his workplace, it would not be the cultural norm to contact female colleagues who are on maternity leave and expresses frustration at the expectation that he continues to engage with the workplace without pay. Therefore, the differential treatment of male and female staff further evidences the role hegemonic expressions of occupational masculinity play in dissuading fathers from taking leave.

Participant F commented on the obvious lack of support for fathers by employers and the obvious gendered division of labour. He commented on the fact that his employer prides itself of being very supportive of women and families by having very generous maternity leave, but at the same time sending the message that fathers should not take extended time off work on leave:


*“My employer makes a big song and dance about how generous the maternity leave policy is where mothers get six months full pay … they are very proud of the fact they sort of support women and support families by having generous maternity leave but at the same time they make it very clear that men effectively should not be taking much time off to take care of their children. I realised there is a weird sort of coheres incentive here that my employer supports women to take time off to take care for children, but does not support men to take time off and take care of children.“*
(Participant F)

However, some fathers in the sample indicate that it is because of their workplace’s culture that they felt empowered to take a more flexible approach to leave. For participant (N), the provision of information and support encouraged him to take SPL, ‘I had already asked before baby came along about shared paternity and what would that entail and how do I go about doing it? And they were very relaxed about it.’

### 3.3. Financial Barriers

Participants indicated that monetary concerns significantly influence their decision-making in relation to paternity leave or SPL. Where the father is the primary earner, current paternity leave and pay provisions can cause the family to experience financial precarity. Therefore, some of the participants chose not to take paternity leave, but instead, to use annual leave allowances that would leave them financially better off:


*“I couldn’t save that kind of extra two- or three-weeks money to take extra time off. So, with the second child I simply took two weeks annual leave… So, in both cases, well in the first case I had statutory paternity leave for two weeks, the second case as far as the business is concerned, I haven’t even had a child I just had two weeks leave.”*
(Participant D)

Participant D alludes to an underlying issue within the structure of parental leave, and the perception of fathers in relation to their female counterparts. It could be argued that some businesses perceive parenthood to be a female experience, and that they do not recognise the impact that it has upon male employees. D’s reflection that he felt as if he ‘hadn’t even had a child’ is indicative of the ways in which fatherhood is under-acknowledged in comparison to motherhood, and of a cultural reinforcement of the division between the public sphere of work and the private sphere of motherhood. Similarly, other participants indicated that they felt neglected or penalized by paternity pay regulations in comparison to their female counterparts.

As Participant F describes, low paternity pay can present a ‘perverse incentive’ for some fathers to forgo their leave in favour of their partners taking more maternity leave. This is compounded where the maternity leave allowance is perceived to be more substantial than the equivalent paternity offering:


*“The more generous maternity leave is, the more there is a perverse incentive for men to not take time off to take care of their kids. Because if women are being offered full pay for six months to take time off work but men are offered nothing for any amount of time to take time off, then the calculation is just sort of very, very quickly done.”*
(Participant F)

Here, Participant F highlights a gendered, structural disparity, and emphasizes that whilst many women receive some financial and occupational support when they take leave, their male counterparts are often subjected to resistance or to punitive arrangements by comparison. For many participants, money acted as a significant stressor, leading fathers to take less time off, or to take different forms of leave, even where employers were more supportive.

Related to SPL, some participants indicated that although the proposal is attractive in principle, it is stifled by the financial difficulties that extended periods of leave would impose upon the family. Many highlight that whilst their employers present SPL as a choice and are supportive, the long-term monetary impact is too severe for them to weather. Participant (W) illustrates this: ”the parental leave wouldn’t really have worked for us because, well especially the financial implication of it, the statutory pay is like a hundred-fifty, two hundred quid a week.”

His reflections also demonstrate the highly unequal gender relations that underpin the obstacles that some families experience.

### 3.4. The Impact of the COVID-19 Pandemic

While most new fathers were not allowed into the birthing suite and they missed experiencing the birth of their baby due to COVID-19 restrictions, most of them valued the opportunity the pandemic gave them to spend more time at home with the baby without the added pressure of physically returning to work, as expressed by participant P, “I just love every moment and I’m really actually incredibly grateful that we have had the pandemic so I can work from home.”

Similarly, Participant G commented on the good relationship he developed with his daughter as a result of being at home and working from home:


*“When the Covid pandemic came in and suddenly everything was turned on its head. In some ways that was quite good for us and for my relationship with my daughter. Because I wasn’t expected to come into [work]. So, I did walks with the baby carrier every day at 11:00 and 16:00 got to know the local geography a lot better than I had previously.”*


Participant R explained that being on the furlough scheme because of the pandemic meant that he could dedicate his time to his child, which would not have been possible if he was working or had gone back to work. He expressed the importance of spending time with his child being more important than going to work and earning money as the bond between him and his child grow stronger:


*“Yeah, we went onto full furlough in the first lockdown, so I didn’t have any work from March until the end of June, until I started the current role that I’m in and it was amazing, I loved it. It was like being retired as I was on full pay. We spent days in the garden, we did all sorts it was—I’d have been happy for that to have been forever that, not having to go to work and getting paid.”*
(Participant R)

Participants felt that working from home gave them the opportunity to be there if they were needed. Participant L was able to help with the other child, which then gave his wife the opportunity to rest more and concentrate on recovering and caring for the newborn baby.


*“Especially with my second child where I was working from home at the time. I’ve been into work once since the start of the pandemic. I’m on hand, you know, during the day if I’m not actually in a meeting with someone, and I’m able then to do my bit with getting the older one to school and everything.”*
(Participant L)

While the pandemic provided an opportunity for fathers to work from home, be present and have more time with the family, some of the participants struggled to combine work and childcare when working from home. Participant S reflected on the challenges that he faced working from home as he struggled to juggle home, life, childcare and work.


*“This was in the early days of the pandemic and then when he was just a few months old we all went into lockdown, so I’ve been essentially with him ever since, right so it’s been great for him. He’s had both his parents around all day every day and it’s been sort of, tough juggling those home, life, childcare, working responsibilities.”*
(Participant S)

## 4. Discussion

The study investigated fathers’ experiences of parental leave entitlement, workplace cultures and the impact of the COVID-19 pandemic on their mental health. While the pandemic may have been challenging to fathers in different ways, such as losing access to informal support networks, fathers were also presented with some ‘unique’ opportunities. The study demonstrates that barriers to leave entitlement, occupational and cultural perceptions of masculinity significantly impact fathers’ ability to spend time with the family when their babies are born. It is important to consider the role of hegemonic masculinity and gendered division of labour that are embedded in the leave available to fathers and the occupational barriers faced by fathers wishing to benefit from them. Although the pandemic was challenging in different ways, lockdown presented an opportunity for fathers to spend more time at home where they would ordinarily have been physically back at work. The pandemic experience was different depending on whether the father could work from home, was furloughed, or went back to work. The study demonstrates that with current leave entitlement, sociocultural and occupational perceptions discourage fathers from taking on contemporary fathering roles, which has been demonstrated to have a positive impact on fathers’ and partners’ mental health and bonding with the baby [45,46].

Hegemonic masculinity and gendered division of labour provides a useful frame through which participants’ experiences of leave entitlement and workplace experiences can be analysed [14]. Gendered perceptions of fathers being the breadwinners of the family and the heterosexual parental division of labour assumptions consider the father to be the mother’s supporter rather than a parent. Although parenting is perceived to have become more egalitarian in the UK, most participants reflected on the engrained cultural assumptions of gendered parental division of labour through leave entitlement and their experiences of taking leave. Gendered division of labour advances the cultural beliefs and perceptions that mothers are primary caregivers in heterosexual families. Most participants commented on how inadequate the two weeks’ paternity leave was and how it undermined their legitimacy as parents and further limited the father’s ability to support the partner or bond with the baby. While policies such as paternity and shared parental leave could increase opportunities for fathers’ involvement in childcare duties, without sacrificing their labour market status [34], the policies largely endorse masculinity norms. Fathers have significantly less leave compared to a mothers’ 52 weeks of maternity leave, endorsing the masculinity norm.

Grounded in hegemonic masculinity perceptions and the perception that fathers are mothers’ supporters rather than parents, most employers support and enhance maternity leave pay but not SPL leave, which fathers could benefit from [67]. These practices are driven by heteronormative assumptions about the gendered division of labour, manifested through the significantly smaller leave entitlement fathers get as opposed to mothers. Also, in terms of the hegemonic masculinity and gendered division of labour perceptions, participants commented on their lack of agency and independence with SPL. Participants felt that their role as parents were not being recognised in policy and participants felt they would be encroaching on the mother’s rights by sharing her maternity leave in the form of SPL. These cultural perceptions of fatherhood prevent fathers from taking up more responsibilities in the family or spending time with the family when the baby is born. The lack of equal treatment from employers where maternity leave is enhanced, but paternity leave or SPL are not, leaves participants unsupported. These hegemonic and gendered divisions of labour perceptions leave fathers feeling unsupported and discouraged from benefiting from leave entitlement that could allow them to be more active parents [68].

The study found that financial cost was one of the key barriers to fathers taking leave. Paternity and SPL are paid at the basic statutory rate, which is significantly lower than the pay of the fathers. While a mother’s maternity leave is paid at 90% of her wages or basic pay, depending on which is higher, paternity leave and SPL are paid at the basic rate. Many of the participants expressed frustration at this gender difference in leave and pay which emphasises the hegemonic masculinity trope that men are the breadwinners of the family. Participants internalise these masculinity norms and tend to take annual leave rather than paternity leave [50]. This feeling was attributed not only to the overwhelming gender divide in support received, but also to practical barriers, such as the expectations that fathers will continue to work while on leave. Participants were frustrated at the fact that employers tended to enhance maternity pay and not paternity or SPL pay, which are barriers that are rooted in heavily gendered dynamics between mothers and fathers. This demonstrates that, in the workplace, fathers are often considered as ancillary or secondary to mothers.

However, merely offering a policy is not sufficient to obtain the desired effect to support fathers [48]. Furthermore, enhancing the pay for the current leave entitlement alone may not equate to greater uptake of parental leave. Instead of sharing the mother’s maternity leave, an increased independent paternity leave entitlement would provide a better opportunity for fathers to take up more active roles in the family and enhance their mental health. Organisational cultures and perceptions need to recognise that men are not just fathers with a role of supporting their partners but are parents and need time to bond with the baby and process the change in routine that comes with fatherhood. Where cultural perceptions of vulnerability and masculinity persist, fathers will continue to internalise their struggles, which will in turn impact on their mental health.

Participants struggled to adapt to the significant change in routine and priorities when the baby was born and talked about two weeks of paternity leave not being enough. Although they could make use of SPL, the financial challenge, and especially the ‘guilt’ of taking the mother’s time, became a significant barrier to having more time off. The lack of time to process the changes in the home and the expectation at work for the father to continue work as normal pose a significant challenge to fathers’ mental health following the birth of the child. While mothers have more time during maternity leave to adapt to changes, fathers do not. Fathers are less likely to have discussions with their employers on the support needed following the birth of their child. Consequently, participants struggled to empathise with themselves and downplayed their struggles by prioritising the mothers’ struggles. Participants only reflected on their wellbeing and mental health after the birth period, giving the impression that they did not prioritise their wellbeing at the point that they were struggling. Participants struggled to navigate fatherhood and their mental health mainly because they tried to fit into the social and cultural construct of masculinity and gender division of labour.

Participants whose babies were born during the COVID-19 pandemic found that although they had to work from home, they felt they had more time with the family than participants who, pre-COVID, had to go into work after their leave [62]. Fathers that were on the furlough scheme had much more time with their family compared to fathers that still had to work from home. While the pandemic presented different challenges, such as the inability to access informal support [63], fathers felt that they did not have to navigate the challenge of negotiating the transition to work. Working from home offered a reason for fathers to be at home and not have to feel guilty or navigate gender and masculine perceptions. Not having to think about navigating expectations and perceptions at work helped fathers’ mental health. However, some of the fathers struggled with working from home and childcare, given that they were in lockdown.

The study suggests that, even though fathers are more eager to take up active roles in the family, fathers’ leave entitlement; organisational and social constructs around hegemonic masculinity; and gender division of labour perception are barriers to fatherhood. These barriers to fathers’ leave entitlement result in a negative impact on the father’s and partner’s mental health and the bond between the father and the child. Gender stereotypes present practical barriers to fatherhood with negative impact on father’s mental health. While fathers’ leave entitlement does not provide sufficient time for a father to bond with the baby, the organisational culture of actively supporting mothers but not fathers present a significant barrier to fathers’ work–life balance and mental health. Although the pandemic lessened the impact of these issues on fathers, there is a need to address the engrained stereotypes around fatherhood that exist pre- and post-pandemic as life returned to normal.

## 5. Conclusions

This study demonstrates that leave entitlement and occupational cultures are key determining factors that shape fathers’ experiences. However, the focus on mothers as primary caregivers has left fathers at a disadvantage in terms of leave entitlement and support offered by the workplace. To fathers, policymakers and employers must recognise the role of a father in the child and mother’s life and make provisions that allow the father to bond with the baby and maintain or improve their wellbeing. Workplaces need to recognise the complexity of fatherhood and embed organisational cultures to support fathers. Predominantly, there is a need for the government to overhaul the current SPL policy and review the paternity leave entitlement to ensure that fathers have an extended, independent, and protected period which would enable them to take up more active roles in the family. This policy change would encourage a wider societal and cultural change in embracing the role of a father as a parent, which would have a positive impact on fathers’ wellbeing. Once policies and workplaces normalise the role of the father in the home, fatherhood barriers driven by gender stereotypes will diminish.

Further research is required to understand the impact of current leave entitlement and occupation cultures on fathers’ sense of achievement. Furthermore, there is a need for further studies to understand how fathers navigate gender stereotypes and attitude towards flexible working requests on fathers’ mental health.

## Figures and Tables

**Table 1 ijerph-20-05454-t001:** Identified themes and subthemes.

Theme	Sub-Themes	Impact on Fathers
Leave entitlement	Two weeks paternity leave being too short. Lack of fathers’ independent right to shared parental leave.	Inability to support the partner or bond with the baby. Inability to adjust to the change in circumstances. Struggle to balance work and family demands at the time. Fathers consider sharing partner’s maternity leave as intrusive.
Occupational and cultural perceptions	Lack of support from employers and colleagues. Lack of information in the workplace.	Leave perceived as encroaching upon work responsibilities. Fathers are discouraged from taking leave. Fathers feel unsupported in taking leave.
Financial barriers	Low pay of paternity and shared parental leave. Employers enhanced maternity leave, but not paternity leave, in shared parental leave pay.	Fathers prefer to take annual leave to minimise financial impact. Fathers perceive the lack of financial support as a signal to not take leave.
The impact of the COVID-19 pandemic	Fathers during lockdown were present and more involved in family activities.	Fathers created more meaningful bonds with their babies and partners.

## Data Availability

Data for the study cannot be made available due to privacy and ethical issues.

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
