# Peer review of "Paternal Leave Entitlement and Workplace Culture: A Key Challenge to Paternal Mental Health"

_ijerph, 2023, doi:10.3390/ijerph20085454_

Round 1

Reviewer 1 Report

The authors explore the compelling and important topic of paternity leave, which may have far-reaching implications for long-term child and family health. The paper is generally well-written and clear. The study design is sound and appropriate, and the findings are interesting and inform policy.

Addressing the following feedback may improve the paper:

Introduction: 

On line 27, what type of diagnosis are the authors referring to? Mental health diagnosis? If so, please specify. 

More importantly, while the goal of the study is to examine how paternal leave policies impact paternal mental health, there is little discussion in the Introduction about findings in the extant literature about this topic. Please expand this discussion in the Introduction.

As paternity leave likely has critical implications for maternal, child, and overall family health and wellbeing, some discussion of these impacts should be included in the Introduction and then referenced later in the Discussion.

There is a typo for instill on line 61.

Materials and Methods:

The sentences in lines 117-121 are not actually sentences—please reword.

Results:

Presenting the main findings in a table format would help the reader better absorb the information.

Discussion:

As stated earlier, please contextualize the implications of your findings for the wellbeing and health of mothers, children, and families in general.

Author Response

Dear Reviewer, thank you for taking the time to read my article and the feedback. I have taken all your feedback on board and amended it as follows (also see the revised manuscript attached):

On line 27, what type of diagnosis are the authors referring to? Mental health diagnosis? If so, please specify. 

Amended 

More importantly, while the goal of the study is to examine how paternal leave policies impact paternal mental health, there is little discussion in the Introduction about findings in the extant literature about this topic. Please expand this discussion in the Introduction.

Literature included in the introduction – see track changes

As paternity leave likely has critical implications for maternal, child, and overall family health and wellbeing, some discussion of these impacts should be included in the Introduction and then referenced later in the Discussion.

Literature added in the introduction on the impact of fathers leave to partner and baby.

There is a typo for instill on line 61.

 Corrected

Materials and Methods:

The sentences in lines 117-121 are not actually sentences—please reword.

 Done

Results:

Presenting the main findings in a table format would help the reader better absorb the information.

Table of themes and sub-themes included in results section

Discussion:

As stated earlier, please contextualize the implications of your findings for the wellbeing and health of mothers, children, and families in general.

Points added in the introduction and discussion sections to cover this

Reviewer 2 Report

I had the opportunity to read an interesting article related to paternal leave. I find the article very interesting and well prepared, but I have a few brief comments for the authors:

- lines 29-37: would you consider transferring this part to the end of the Introduction?

- lines 142-145: would you consider moving this part to the Institutional review board statement?

- line 146: please consider renaming the chapter into Results

- line 148: you are mentioning three themes here but in the further text you describe four of them - please unify the statements

- Discussion: could you please provide some of the limitations to your study?

- the  article is missing supplemental information (Supplementary materials, Author Contributions, Funding, Institutional review board statement, Informed consent statement, Data availability statement, Acknowledgments, Conflicts of interest)

Author Response

Dear Reviewer, thank you for taking the time to read my article and provide feedback. I have taken all your helpful comments on board and addressed as follows (please see revised manuscript attached as well):

- lines 29-37: would you consider transferring this part to the end of the Introduction?

Moved to the end of introduction

- lines 142-145: would you consider moving this part to the Institutional review board statement?

Moved

- line 146: please consider renaming the chapter into Results

Renamed

- line 148: you are mentioning three themes here but in the further text you describe four of them - please unify the statements

corrected

- Discussion: could you please provide some of the limitations to your study?

Limitations were included in lines 136-140

- the  article is missing supplemental information (Supplementary materials, Author Contributions, Funding, Institutional review board statement, Informed consent statement, Data availability statement, Acknowledgments, Conflicts of interest)

Included

Reviewer 3 Report

Dear authors,

Thank you for the opportunity to review your paper, that addresses important gender and cultural aspects on the topic paternity leave and modern family life.

I have some comments that I believe would improve your paper:

- Throughout the paper, there are some issues with language and spelling, that you should get sorted out.

- Introduction: How is the financial aspect visible? In many countries, there is a  pay gap between men and women –  does this or other financial aspects have an effect?

- Introduction: It would be useful with more information/research about the effects of covid on parental leave/families, since covid is one of the main topics in the interviews

- Introduction: You could focus more on the barriers - which specific barriers have been found in previous studies?

- Lines 40-42: What is masculinity, masculine norms and fatherhood ideals? A brief discussion of the topics would be good here.

- Line 47: Traditional gender norms: more references are needed here

- Line 55: Which government? Maybe address the fact that there are very different conditions or paternal leave around the world

- Lines 57-58: By whom? And how is this evident?

- Line 63: What about the child´s perspective?

- Line 82: Spell out SPL here

- Line 129: What kind of thematic analysis? And please add references for this

- Lines 379-381: Gender and cultural perceptions: is this evident in your data?

- Line 469: Experiences of what?

Author Response

Dear Reviewer, thank you for taking the time to read my article and for the helpful comments. I have taken all the comments on board and addressed them as follows (please also see attached revised manuscript):

- Throughout the paper, there are some issues with language and spelling, that you should get sorted out.

Proof read and corrections made

- Introduction: How is the financial aspect visible? In many countries, there is a  pay gap between men and women –  does this or other financial aspects have an effect?

Literature included

- Introduction: It would be useful with more information/research about the effects of covid on parental leave/families, since covid is one of the main topics in the interviews

Literature added in the introduction

- Introduction: You could focus more on the barriers - which specific barriers have been found in previous studies?

 Parts of introduction rewritten and literature added

- Lines 40-42: What is masculinity, masculine norms and fatherhood ideals? A brief discussion of the topics would be good here.

Added – see attached corrected manuscript

- Line 47: Traditional gender norms: more references are needed here

Connell 1990 added

- Line 55: Which government? Maybe address the fact that there are very different conditions or paternal leave around the world

I have specified that it's UK. Respectfully, I don’t find it necessary to discuss the international context because the paper does not have the capacity to do justice to that aspect currently. But this is definitely an angle to explore in another paper in its own right

- Lines 57-58: By whom? And how is this evident?

Clarified using the disproportionality in leave entitlement between a mother and a father

- Line 63: What about the child´s perspective?

The impact of the father’s presence on the child added

- Line 82: Spell out SPL here

Done

- Line 129: What kind of thematic analysis? And please add references for this

Reference added

- Lines 379-381: Gender and cultural perceptions: is this evident in your data?

Yes, e.g. lines 308-319 

- Line 469: Experiences of what?

Transitioning into parenthood added

Round 2

Reviewer 1 Report

The author has satisfactorily addressed all recommendations. Please note that the word "mental" is misspelled on line 94, and the sentence on lines 96-97 needs to be amended as it does not currently make sense. 

Author Response

Dear Reviewer,

Thank you for taking the time to read and provide me with feedback.

I have corrected line 94 to read 'mental'.

I have amended lines 96-97 to read 'Research demonstrates that when fathers take leave, their availability and support improve the mental health of their partners...'

Thank you.